# An Intermittent Fasting Mimicking Nutrition Bar Extends Physiologic Ketosis in Time Restricted Eating: A Randomized, Controlled, Parallel-Arm Study

**DOI:** 10.3390/nu13051523

**Published:** 2021-04-30

**Authors:** Angie W. Huang, Min Wei, Sara Caputo, Melissa L. Wilson, Joseph Antoun, William C. Hsu

**Affiliations:** 1L-Nutra, Inc., Plano, TX 75024, USA; awu@l-nutra.com (A.W.H.); mwei@l-nutra.com (M.W.); scaputo@l-nutra.com (S.C.); jantoun@l-nutra.com (J.A.); 2Preventive Medicine, Keck School of Medicine, University of Southern California, Los Angeles, CA 90089, USA; melisslw@usc.edu

**Keywords:** diet, intermittent fasting, time-restricted eating, fasting ketone, breakfast, intermittent fasting bar

## Abstract

There has been increasing interest in time-restricted eating to attain intermittent fasting’s metabolic benefits. However, a more extended daily fast poses many challenges. This study was designed to evaluate the effects of a 200-calorie fasting-mimicking diet (FMD) energy bar formulated to prolong ketogenesis and mitigate fasting-associated side effects. A randomized, controlled study was conducted comparing the impact of consuming an FMD bar vs. continued water fast, after a 15-h overnight fast. Subjects in the FMD group showed a 3-h postprandial beta-hydroxybutyrate (BHB) level and 4-h postprandial BHB area under the curve (*AUC*_0–4_) that were non-inferior to those who continued with the water fast (*p *= 0.891 and *p *= 0.377, respectively). The postprandial glucose *AUC*_0–4_ in the FMD group was non-inferior to that in the water fast group (*p *= 0.899). A breakfast group served as a control, which confirmed that the instrument used in home glucose and ketone monitoring functioned as expected. The results indicate that FMD bar consumption does not interfere with the physiological ketogenesis associated with overnight fasting and could be used to facilitate the practice of time-restricted eating or intermittent fasting.

## 1. Introduction

Reducing caloric intake without malnutrition is a robust intervention to extend lifespan in model organisms and impart health benefits in humans [1,2,3,4]. Due to the potential adverse effects associated with chronic dietary restriction and the challenges in following diet programs that affect the balanced intake of macronutrients and healthy foods such as vegetable, fruits, nuts and whole-grain cereals [5], new nutrition research has been focusing on intermittent fasting (IF). IF explores the fundamental biochemical mechanisms underlying the fasting-refeeding cycles that elicit defenses against oxidative and metabolic stress, promote cell/tissue repair during fasting and growth and rejuvenation during refeeding [6,7,8,9,10].

After a meal, glucose is used for energy or converted for storage as glycogen or triglyceride. Fasting leads to the mobilization of hepatic glycogen stores. As the hepatic glycogen stores are being spent (after 8–12 h of water-only fasting), the liver starts to produce ketones, i.e., acetoacetate and ß-hydroxybutyrate (BHB), to provide an energy substrate for peripheral tissues. Blood BHB level could reach 2 mM after 48 h and keep rising as the fasting continues. BHB is not only a metabolic intermediate but also acts as a signal to regulate metabolism during nutrient deprivation and modulate signaling pathways and cellular functions implicated in human diseases and aging [11,12]. The timing and mechanism of fasting-induced ketogenic response gives rise to several popular intermittent fasting regimens, such as time-restricted eating (e.g., 16-h fasting per day or 16:8 TRE) [13,14], alternate day fasting [15,16,17], the 5:2 fasting [18], and fasting-mimicking diets [19]. Many of these eating patterns had been shown to impart health benefits, including weight control, improved glucose metabolism, ameliorated cardiovascular risk factors, augmented cancer outcomes, and supporting healthy aging [8,9,10,15,19,20,21,22,23]. However, it is not clear if the observed benefits are solely due to energy reduction, the timing of fasting vs. refeeding, the restriction on specific macronutrients, or the combination of these factors [24].

Extending the period of fasting as part of IF, however, can be difficult for certain practitioners. Changing food habits, hunger, and interfering with social interactions are among the barriers that hinder the adherence to such dietary interventions [25]. Individuals following an IF diet tend to skip breakfast and save their eating for dinner or a late evening snack [26]. Meta-analyses had shown that skipping breakfast is associated with an increased risk of type 2 diabetes [27]. Such association and significantly increased risk of mortality from cardiovascular disease was also observed in a long-term prospective study on meal skipping [28,29]. Prolonged abstinence from food may result in decreased eating control (including increased portions of unhealthy foods and binge eating) during the non-fasting period [25] and may also lead to increased lithogenicity in bile and potential gallstone disease [30,31,32,33]. To alleviate the burden and minimize the potential adverse effects of intermittent fasting as well as behavioral barriers driven by hunger and the lack of food, we tested a specially formulated energy bar designed based on the previously described fasting-mimicking diet (FMD) [19,34,35,36,37]. Through technology such as the fasting mimicking diet, we are exploring the potential of novel diet formulation to achieving many of the benefits of fasting while avoiding some of the challenges associated with meal skipping.

## 2. Materials and Methods

### 2.1. Participants

Participants for this dietary intervention study (Clinical Trial Registry: NCT04499599) were recruited via email and web advertisement from July to September 2020 under the protocol approved by the Institutional Review Board of WIRB-Copernicus Group (Protocol #20201920). The study was conducted remotely during the COVID-19 pandemic period through phone and video conference calls. The study information was gathered and managed with electronic clinical trial management software (CTMS, ClinicalResearch.io, Boston, MA, USA). All participants gave their informed consent for inclusion before they participated in the study. Eligibility criteria for the study included age between 18 and 65 years at screening, ability to provide informed consent, body mass index (BMI) between 20 and 35, ability to complete assessments in English, e-mail and internet video conference access, in general good health, and willingness to consume all foods served in the protocol. Exclusion criteria included history of gastric bypass, type 1 diabetes, on medications aimed at keeping blood glucose under control, allergies to study foods, pregnancy and alcohol dependency.

### 2.2. Study Design

This is a randomized, controlled, parallel-arm study that evaluates the impact of a test formulation, FMD bar, on ketogenesis and satiety as compared to a 19-h water-only fast. As the study was performed during the COVID-19 pandemic period where conducting conventional university-based clinical trials was not possible, the adoption of virtual technology and methodologies presented a novel approach to assessing formulation’s performance and obtaining user-reported outcomes. The glucose and ketone were measured with a commercial glucose/ketone meter through a finger stick instead of phlebotomy, a breakfast group was included as a control to ensure that the instrument functioned as expected. The breakfast was in the form of a bar for easier preparation and shipping during the pandemic. The breakfast composition (Table 1) was determined based on NHANES (2013–2014) and published data [38]. Due to the typical nutrient composition of a breakfast, we anticipated a rise of glucose level and a decline in ketone level immediately following ingestion. Eligible participants were block-randomized (12 blocks of 9 each) in a 1:1:1 allocation to the breakfast, Fast Bar (FMD bar) and water-only fast groups using a predetermined randomization table (http://www.randomization.com accessed on 6 June 2020). Study materials, including a digital scale, study foods (Table 1), a Precision Xtra Blood Glucose & Ketone Monitoring System (Abbott) and test supplies as well as instruction materials, were sent to participant’s home. On the day 1 of the study, subjects in all groups were asked to consume a standardized ready-to-eat dinner meal at 5 pm and, then, fast overnight for approximately 15 h. On day 2, subjects in water-only fast, FMD, and breakfast groups were asked to continue the water-only fast for 4 more hours, to consume a FMD bar or a breakfast bar, respectively. Blood ketone (BHB) and glucose levels were assessed at baseline and hourly after consuming study foods for 4 h. Subjects were also asked to complete a self-administered online survey (https://www.clinicalresearch.io/), including self-reported height and weight (on study provided scale) as well as information regarding demographics, medical history, concomitant medications, physical activity, drinking habits (all modeled on the Nurses’ Health Study questionnaires). Besides, participants were asked to answer questions on food quality, satiety, and mood (nervous, blue, annoyed, fatigued, restless, cheerful, relaxed, in control, refreshed, rejuvenated) on a Likert Scale, before and 3 h after consuming the study foods to assessing subjective endpoints related to the study foods. The primary outcomes were 3-h postprandial ketone and glucose levels (day 2) and ketone and glucose Area Under the Curve (AUC) for the 4-h period following the breakfast (general time frame until the next meal). A non-inferiority test was conducted comparing the FMD bar vs. water fast on the primary outcomes.

### 2.3. Blood Glucose and Ketone

Blood glucose and ketone readings were assessed after 15-h overnight water-only fast (baseline) and hourly after consuming study foods for 4 h using the Precision Xtra Blood Glucose & Ketone Monitoring System (Abbott), a widely used point-of-care kit which had been validated against standard laboratory method in animals and humans [39,40,41,42,43,44].

### 2.4. Statistical Analysis

Sample size estimation was based on pilot studies assuming ketone (BHB) levels of 0.25 (SD: 0.19) and 0.16 (SD: 0.05) mM for the Fast Bar and the Breakfast groups, respectively. With t test (one-tailed) of difference between two independent means, a total *n* = 31 will be necessary to achieve 80% power with an α error probability of 0.05 (GPower v3.1.9.6, Franz Faul, University Kiel, Germany). A 10% dropout rate was anticipated based on prior experience in similar dietary intervention studies. Demographic and clinical characteristics were described according to study group using means and standard deviations for continuous variables and counts with frequencies for categorical variables. To examine the impact of the intervention on glucose and BHB at each time point, we used one-way ANOVA with post hoc pairwise comparisons and adjustment for multiple comparisons using Tukey’s method. Non-inferiority was defined based on a power of 80% (β = 0.2), type 1 error (alpha at two-sided 95% confidence intervals) of 0.025, and a non-inferiority margin of 15% for the relative mean difference. Non-inferiority was assessed with a Student’s *t*-distribution and evaluated whether the means for the 3-h postprandial ketone level on day 2 and area under the curve (*AUC*_0–4_) for BHB within the FMD bar (new treatment method) group was no more than a 15% difference in comparison to the water fast (active comparator) group. To further evaluate the impact of the intervention on glucose levels over time, a linear mixed effects model was built, including treatment arm and adjusting for age, activity level, BMI category (normal, overweight, obese) and sex. A random effect was specified for the participant. We tested the following interactions with the treatment arm in all models: sex, age, BMI category, and activity level. We subsequently estimated marginal means for each of the variables in the model and tested for significant differences, adjusting for multiple comparisons using Scheffe’s method. Variables were included in the model only if they were statistically significant predictors, confounders, or interactions. A confounder was defined as any variable that altered the relationship between treatment arm and study endpoint by >15%. Model fit was assessed by plotting residuals and evaluating deviation; all models were well-fit. An adjusted *p*-value < 0.05 was considered significant. Analyses were conducting using Stata 16.0 (StataCorp, College Station, TX, USA).

## 3. Results

### 3.1. Baseline Characteristics

Of 110 potential subjects who responded, 1 declined to participate, 3 were deemed ineligible due to missed study appointment, excess BMI, and pregnancy, and 1 was disqualified due to SARS-CoV-2 positive diagnosis (Figure 1). At the start of the study, the three groups were similar in age, weight and BMI (Table 2). There were more female participants than male participants (77.1% vs. 22.9%) with the water-only fast group having the most under-represented male participants.

### 3.2. Blood Beta-Hydroxybutyrate (BHB) and Glucose Levels

At baseline, no differences were noted in blood glucose and BHB levels between groups (Table 3). Blood BHB level showed a steady increase in the water-only fast group, whereas the BHB level dipped below the baseline in the breakfast group, recovering 4 h postprandial (Figure 2a). The FMD bar group showed a steady blood BHB increase 1 h after food consumption and reached the same level of that water-only fast group 3 h post FMD consumption (Figure 2b and Table 3). The mean 3-h postprandial ketone levels on day 2 were 0.369 (0.207) mM and 0.374 (0.130) mM for the water fast and FMD bar groups, respectively. The mean difference between the water fast and FMD bar groups was 0.006 (95% CI, −0.089 to 0.077) mM. The difference was above the lower bound 95% CI for the non-inferiority margin (−15%), demonstrating the FMD bar group is non-inferior to the water fast group in terms of ketone levels at hour 3 after consuming food (relative mean difference = 1.35%). The water fast group mean was 1.267 (0.529) mM∙h and the FMD bar group mean was 1.172 (0.336) mM∙h for the area under the curve (*AUC*_0–4_) for BHB with a mean difference of −0.094 (95% CI, −0.117 to 0.305) mM∙h (Table 4). The mean difference was above the lower bound 95% CI for the non-inferiority margin (−15%), demonstrating the FMD bar group is non-inferior to the water fast group in terms of *AUC*_0–4_ for BHB (relative mean difference = −7.50%).

There was no significant difference in blood glucose levels between the study groups at 3 h after consuming study foods (Table 3). The water fast group mean was 365.0 (50.0) mg/dL∙h and the FMD bar group mean was 363.5 (37.3) mg/dL∙h for the glucose *AUC*_0–4_. The mean difference between the water fast and FMD bar groups was −1.47 (95% CI, −19.55 to 22.50) mg/dL∙h. The difference was above the lower bound 95% CI for the non-inferiority margin (−15%), demonstrating the FMD bar group is non-inferior to the water fast group in terms of the area under the curve (*AUC*_0–4_) for glucose (relative mean difference = −0.41%).

Lastly, we modeled *AUC*_0–4_ for both glucose and BHB using linear regression with assessment for confounding and interactions. We found no significant predictors or interactions and we did not find evidence of confounding. Thus, all models are univariate (data now shown).

### 3.3. Subjective Endpoints

Participants completed a self-administered online questionnaire including 4 questions on satiety, 7 on food quality, and 10 regarding mood. Continued abstinence from food after a 15-h overnight fast led to decreased feeling of fullness and increased desired to eat. FMD received similar flavor/quality rating as the breakfast (Appendix A). FMD bar significantly imparted fullness and curbed the desire to eat 3 h after the consumption of the study foods compared to water fast (Figure 3). No difference was observed in the mood questionnaire comparing the FMD bar and water fast (Appendix A).

## 4. Discussion

This study aimed to evaluate how a specially formulated fasting-mimicking nutrition bar affects fasting physiology and subject-reported outcomes following an overnight fast. FMD bar consumption led to similar levels of 3-h postprandial blood ketone and ketone *AUC*_0–4_ as those of water-only fast. Moreover, postprandial glucose *AUC*_0–4_ of the FMD group was comparable to that of the water fast. The elevation in glucose and the decline in ketone in the breakfast control confirmed that the instrument used to measure ketone and glucose levels functioned as expected. The FMD bar, despite containing nutrients, did not elicit a 1-h postprandial glucose elevation in contrast to the breakfast. Our results demonstrated that manipulation of energy intake and macronutrient composition can extend physiological ketogenesis similar to water fast while providing some energy and nutrients to increase the sense of fullness and satiety. Another novel aspect of the study was the rapid and simple way the study was conducted remotely during the pandemic to test foods that extend the fasting-associated ketogenesis.

Discoveries linking the nutrient-sensing pathway and aging led to the development of novel fasting-based dietary interventions that downregulate pro-growth signaling, activate cellular protection, and ameliorate risk factors associated with diseases and aging. Although intermittent fasting or time-restricted eating patterns gained interest among fasting practitioners in recent years, it could still be challenging to implement due to fatigue, hunger, potential gallstone disease, and other adverse effects [25,30,31,32,33]. To this end, a fasting-mimicking diet was developed that is able to mimic the effects that water-only fasting has on IGF-1, IGF binding protein 1 (IGFBP-1), ketone bodies, and glucose [19,36,37,45]. This study further demonstrated that the FMD bar, designed with reduced calories, carbohydrates and protein composition, when consumed after an overnight fast, could extend the physiological ketogenesis of the fast without eliciting a postprandial glucose elevation associated with consuming a typical breakfast.

One major difference between FMD technology and a ketogenic diet food (e.g., keto bars) is the percentage of calories derived from carbohydrates and protein. Keto based energy bars primarily consist of high fat (60–90% of total energy), moderate protein (10–30% of total energy), and very low carbohydrates (5–10% of total energy) [46], whereas FMD bar contains lower protein composition and moderate complex carbohydrates (Table 1). Low protein intake had been shown to be more effective in lowering the IGF-1/mTOR nutrient-sensing pathway and is also associated with a reduction in cancer and mortality in the 65 and younger population [47] and protein intake is a key determinant of circulating IGF-1 levels in humans [48]. Furthermore, moderate carbohydrates intake may also spare the gluconeogenesis using amino acids released from muscle protein, thus preserving lean body mass. Administration of as little as 7.5 g carbohydrate to an otherwise starving human halves urinary nitrogen loss [49]. It is important to note that the benefits of protein restriction associated with FMD were in the context of a short-term fast followed by a refeeding period [19,45], in contrast to a long-term protein restriction resulting from chronic nutrition deprivation.

Time-restricted eating (TRE) refers to the daily practice of consuming food within 6–8 h (i.e., 16:8 or 18:6 TRE). We showed that, despite an initial (1-h postprandial) dip of ketone level in the FMD group, the 4-h ketone and glucose AUCs were similar to those of water-only fast after a 15-h overnight fast (Figure 1). These results demonstrated that the FMD bar after an overnight fast could keep the body in a physiologic fasting state comparable to 18 to 19-h water fasting. It not only provides nutrients but may improve adherence, in terms of the feeling of satiety (Figure 3), to the time-restricted eating pattern of 16:8 or 18:6. The fasting-mimicking diet technology opens up the possibility of consuming food with certain calories and macronutrient to achieve some of the benefits of a water-only fast while preserving physiologic (i.e., ketogenesis or glucose homeostasis) or molecular mechanisms associated with fasting. Therefore, while the concept of fasting is generally understood as the absence of food intake, an emerging concept such as physiologic or molecular fasting can be further supported with our study.

### Limitations

During the COVID-19 pandemic, this trial was carried out remotely in participants’ homes instead of a clinical trial center. Despite this limitation, it is important to note that the design was possible to execute under pandemic conditions, with highly efficient enrolment (106 randomized out of 110 assessed for eligibility) and 99% of randomized subjects (105 out of 106) completing the trial. With no travel to clinical centers necessary and brief and straightforward procedures for subjects to follow, the dropout rate was minimal. Through live video and phone calls and electronic data capture technology, the study staff conducted the informed content, provided glucose/ketone testing instructions, and efficiently guided the participants through the study procedures. This approach can be considered for evaluating other foods in this context.

As having phlebotomy and other clinical tests at home were not practical, the effects of FMD bar on physiological pathways that are influenced or modified by intermittent fasting were not evaluated. Future in-clinic study could evaluate the impact of FMD bar on the nutrient sensing pathways and other important biomarkers associated with extended fasting. Since no postprandial glucose elevation was observed after FMD bar intake, it will be interesting to see whether postprandial insulin response is also attenuated and how IGF-1 related system would respond [50]. Another limitation of the study is the disproportional large female participation (Table 2). Estrogen, among other hormones, regulates glucose metabolism [51]. Future studies with more balanced sex ratio and detailed information on hormone profile (e.g., menopause and hormone therapy status) could allow a better understanding how age and sex confound the effects of TRE. Lastly, longitudinal studies incorporating an FMD bar as part of an intermittent fasting or time-restricted eating program are warranted to evaluate its impact on promoting adherence and added health benefits in light of the potential association between meal skipping and increased risk of type 2 diabetes and cardiovascular events [27,28,29]. Of particular interest is the subjective assessment of satiety and mood. The FMD bar group had less desire to eat and experienced less reduction in fullness from the baseline than the water-only group. The outcome of the satiety assessment was limited because the evaluation was performed only 3 h after subjects consumed the study foods. The gap period probably should be in line with the consideration that the FMD bar could function as a snack or in place of skipping a meal, thus be tested after a shortened (e.g., 1 h) wait period, in addition to the 3-h mark to assess the full impact over a time period [52]. The behavioral implication of this finding should be tested in a longitudinal study with more sensitive and validated instruments, including the ones on mood, in the future. With further study, the consumption of the foods with particular energy content and macronutrient ratios may represent a healthy alternative to skipping breakfast to prolong the intermittent fasting period.

## 5. Conclusions

Among generally healthy adults, FMD bar consumption does not lead to postprandial glucose elevations and extends ketogenesis associated with overnight fasts. These results demonstrated that the consumption of foods with a particular energy and macronutrient composition such as a fasting-mimicking diet bar could be a viable strategy to extend the physiological ketogenesis associated with fasting, while providing some nutrients to the body and imparting satiety, to support the practice of time-restricted eating.

## Figures and Tables

**Figure 1 nutrients-13-01523-f001:**
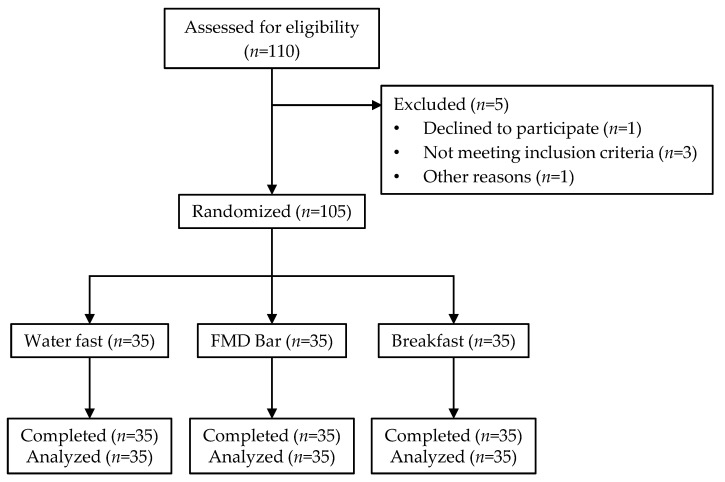
CONSORT diagram of 105 subjects participating in the FMD (fasting-mimicking diet) bar study (July to September 2020).

**Figure 2 nutrients-13-01523-f002:**
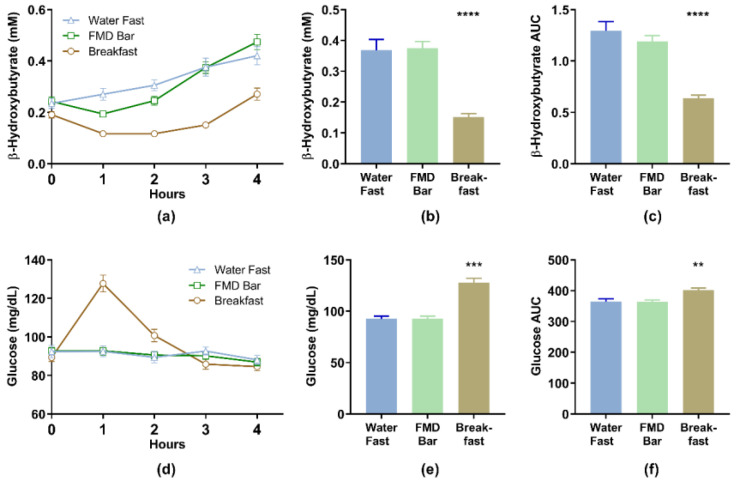
After 15 h of overnight fast, study foods were consumed at hour 0 for the breakfast and FMD groups. (**a**) Ketone and (**d**) glucose levels were measured at time 0 and hourly for 4 h. (**b**) Blood ketone levels 3 h after the experimental groups consumed the study foods. (**c**) Ketone and (**f**) glucose area under the curve (AUC) between hour 0 and 4 after the experimental groups consuming the study foods. (**e**) Blood glucose levels 1 h after the experimental groups consuming the study foods. Data represent mean and standard error. ** *p* < 0.01, *** *p* < 0.001, **** *p* < 0.0001, FMD or breakfast group compared to the water fast group, one−way ANOVA, with Tukey’s multiple comparisons test.

**Figure 3 nutrients-13-01523-f003:**
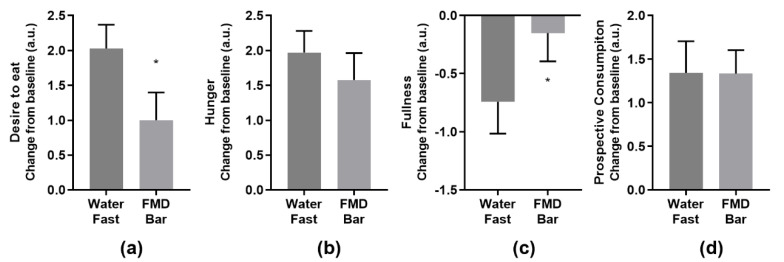
Changes in subjective measures at 3 h post FMD bar consumption: (**a**) “How strong is your desire to eat?”; (**b**) “How hungry do you feel?”; (**c**) “How full do you feel?”; (**d**) How much food do you think you could eat?”. * *p* < 0.05, unpaired, nonparametric, one−tailed, Mann−Whitney test comparing FMD group to the water fast group.

**Table 1 nutrients-13-01523-t001:** Macronutrient composition of the study foods.

	Ready-to-Eat Dinner ^1^	FMD Bar	Breakfast
Weight (g)	662	40	110
Energy (kcal)	660	200	435
Protein (g) (% calorie)	55 (33%)	4.6 (9%)	16.9 (16%)
Fat (g) (% calorie)	25 (34%)	17.3 (77%)	13.5 (28%)
SaFat (g)	3.5	2	2
Carbohydrate (g) (% calorie)	54 (33%)	7.3 (14%)	61.4 (56%)
Sugar (g)	13	5	28
Fiber (g)	8	7	5

^1^ Okinawa Pescatarian Dinner: salmon teriyaki with basmati rice and Asian stir fry vegetables (provided by Nutrition For Longevity, Hackettstown, NJ, USA). FMD, fasting-mimicking diet.

**Table 2 nutrients-13-01523-t002:** Baseline sample characteristics. Continuous variables are described using mean ± standard deviation, while categorical variables are described with count (frequency).

Characteristics	All (*N* = 105)		Arm 1 (*n* = 35)		Arm 2 (*n* = 35)		Arm 3 (*n* = 35)	
			Water Fast		FMD Bar		Breakfast	
Sex, *n* (%)	105		35		35		35	
Female	72	(68.6)	27	(77.1)	23	(65.7)	22	(62.9)
Male	33	(31.4)	8	(22.9)	12	(34.3)	13	(37.1)
Race or ethnic group, *n* (%)								
White	82	(78.1)	25	(71.4)	30	(85.7)	27	(77.1)
Black	6	(5.7)	2	(5.7)	1	(2.9)	3	(8.6)
Hispanic	12	(11.4)	7	(20)	3	(8.6)	2	(5.7)
Asian	5	(4.8)	1	(2.9)	1	(2.9)	3	(8.6)
Age (years)	46.1	±11.6	46.0	±11.5	46.4	±10.9	45.9	±12.5
Weight (kg)	71.9	±12.8	67.7	±11.1	73.0	±12.3	75.0	±13.9
BMI	25.0	±3.5	23.6	±2.9	25.4	±3.4	25.9	±3.8

**Table 3 nutrients-13-01523-t003:** Beta-hydroxybutyrate (BHB) and glucose levels.

	β-Hydroxybutyrate (mM)					Glucose (mg/dL)				
	Mean	SD	95% CI	*p* ^1^	*p* ^2^	Mean	SD	95% CI	*p* ^1^	*p* ^2^
**Baseline**										
Water Fast	0.25	0.15	0.20–0.30			92.4	18.9	85.9–98.9		
FMD Bar	0.24	0.10	0.21–0.28	0.616		92.7	10.8	89.1–96.4	-	
Breakfast	0.19	0.07	0.17–0.22	0.254	0.032	89.3	12.4	85.1–93.6	-	-
**Hour 1**										
Water Fast	0.27	0.14	0.22–0.31			92.5	16.5	86.9–98.2		
FMD Bar	0.19	0.06	0.17–0.22	0.005		92.8	14.6	87.8–97.9	0.998	
Breakfast	0.12	0.05	0.10–0.13	<0.001	0.002	127.8	25.8	118.9–136.6	<0.001	<0.001
**Hour 2**										
Water Fast	0.30	0.13	0.26–0.34			89.43	17	83.5–95.3		
FMD Bar	0.25	0.10	0.21–0.28	0.061		90.63	11	86.9–94.4	-	
Breakfast	0.12	0.05	0.10–0.13	<0.001	<0.001	100.7	19	94.3–107.2	-	-
**Hour 3**										
Water Fast	0.37	0.21	0.30–0.44			92.7	12.3	88.5–97.0		
FMD Bar	0.37	0.13	0.33–0.42	0.986		90.2	10.8	86.5–93.9	-	
Breakfast	0.15	0.07	0.13–0.18	<0.001	<0.001	85.9	15.7	80.5–91.3	-	-
**Hour 4**										
Water Fast	0.41	0.21	0.34–0.49			88.1	13.7	83.4–92.8		
FMD Bar	0.47	0.17	0.41–0.53	0.331		86.9	11.4	83.0–90.9	0.917	
Breakfast	0.27	0.14	0.22–0.32	0.003	<0.001	84.6	12.8	80.2–89.0	0.481	0.726
***AUC*** **_0–4h_**										
Water Fast	1.27	0.53	1.09–1.45			365.0	50.0	347.8–382.1		
FMD Bar	1.17	0.34	1.06–1.29	0.485		363.5	37.3	350.7–376.3	0.989	
Breakfast	0.62	0.18	0.56–0.68	<0.001	<0.001	401.4	42.3	386.9–415.9	0.002	0.001

^1^ FMD bar or breakfast compared to water fast; one-way ANOVA with Tukey’s multiple comparisons test showing only post hoc analyses for significant omnibus F test. ^2^ Breakfast compared to FMD bar, one-way ANOVA, with Tukey’s multiple comparisons test showing only post hoc analyses for significant omnibus F test.

**Table 4 nutrients-13-01523-t004:** Student’s t distribution for mean difference for non-inferiority margin.

	t	df	*p*	Mean Difference	95% CI	
					Lower	Upper
BHB (hour 3)	0.137	58.2	0.891	0.006	−0.089	0.078
Glucose (*AUC*_0–4_)	−0.140	68.0	0.889	−1.471	−19.55	22.50
BHB(*AUC*_0–4_)	−0.889	68.0	0.377	−0.094	−0.118	0.307

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
