# Peer review of "An Intermittent Fasting Mimicking Nutrition Bar Extends Physiologic Ketosis in Time Restricted Eating: A Randomized, Controlled, Parallel-Arm Study"

_nutrients, 2021, doi:10.3390/nu13051523_

Round 1

Reviewer 1 Report

Major comments:

This is generally an informative study and the manuscript is well written. I have one major comment.  

  1. Subjective responses are integral to the interpretation of the study. This is evidenced by the focus on these outcomes in the introduction and discussion. Whilst I appreciate that the subjective outcomes were not the primary focus of the investigation, nevertheless, these assessed subjective outcomes warrant greater focus than is currently present. Please add details of what the specific end-points are and how these end-points were assessed to the methods. In addition, please provide the full data set in the results (i.e. figure 3). This includes the breakfast (control), FMD and water-only trials.

I have also provided several line-by-line minor comments that I hope the authors find useful.

Minor comments

Abstract:

Line 24: Add a p-value for this effect

Reframing of the study as water vs FMB is better.

Introduction:

Line 44-49: This is a very long sentence. Consider splitting this into two separate sentences.

Line 68: Given the focus on appetite and compensatory eating discussed in the previous paragraph, this should be highlighted as a study outcome in this paragraph. This is an important part of the rationale for the study.

Methods:

Line 88: This line is confusing as the it mentions the breakfast condition in the randomisation schedule, as this study has been altered from a three trial to a two trial study. As such, it may be sensible to include lines 92-100 earlier, so that the purpose of the breakfast condition is clear to the reader.

Line 109: I’m not sure why ‘or’ is used in this sentence. Consider revising.

Table 1: Please include the grams for each macronutrient (fat, protein, carbohydrate) in the table.

Table 1: The control breakfast should be included in this table.

Line 120: How was the power calculation performed? Was this related to the FMD vs. water comparison or the FMD vs. control breakfast comparison?

Line 133-137: The measurement of these individual factors is not included in the manuscript. For example, were weight and height self-reported or independently measured by researchers? In particular, how physical activity level was determined is not explained. Please include how all of these individual factors were measured in the manuscript.

Line 150: Reword this sentence to ‘outnumbered males more than 2-fold’

Line 171-188: Please add units here, as appropriate. It may be more efficient to describe in the statistical analysis section of the methods that data is presented as mean (sd), to avoid the need to report (SD =) in each case here.

Line 190-191: Check this sentence for errors.

Line 198-204: These endpoints are not fully detailed in the methods, but are important for the interpretation of the study. This is confusing as the importance of appetite regulation in the introduction, yet the details pertaining to this in the methods and results of the manuscript are minimal. This is in stark contrast to the focus on (fairly predictable) effects of a very-low carbohydrate snack on glucose and BHB concentrations. Please add these details to the methods.

Figure 3: Present both baseline and 3-hour data points in this figure to observe the effect more clearly. It might also be interesting to include the breakfast meal in this figure for context. Add all the subjective outcomes to this figure. Details on exactly how these end-points were assessed (scale, anchors etc.) is needed in the methods.

Discussion:

Line 230: The low carbohydrate content of the FMD bar is the major factor driving the observed effects and this should be highlighted in this sentence.

Line 224-237: Where have you derived the criteria for the descriptors? What constitutes high, moderate, low and very low in regard to macronutrient composition?  

Line 237-239: This is very selective and focuses on the possible negative effects of protein. For balance, the potential positive effects of protein should also be highlighted here. In addition, are these mechanistic effects due to acute protein restriction? What would happen with subsequent feeding?  

Line 239-242: The value stated for glucose is very specific and unlikely to be true for every person. The type of carbohydrate provided may also influence this response. Linking to the previous sentence, this sentence is designed to explain how carbohydrate provision reduces muscle-derived gluconeogenesis from amino acids, which could also be achieved via protein provision in the diet. These sentences read very selective and convoluted, and need consider the whole diet and wider implications of such an intervention.

Line 249-254: This sentence is very good and identifies clearly the application of your findings. A major component of this is the potential to consume food and avoid some of the challenges associated with fasting. This needs to be a greater focus throughout the manuscript.

Line 280-283: Whilst I agree, a more detailed hourly subjective profile would be very beneficial to track the temporal response, this sentence is contradictory as it is stated that FMD is a snack, yet could be viewed as a healthy alternative to skipping breakfast (a meal). In this regard, if the FMD bar is designed to completely replace breakfast, the 3-hour time interval for the measurement of subjective responses is relevant, as practitioners would be expected to maintain the fast for the same duration as if eating breakfast. 

Author Response

We thank the reviewer’s supportive comments and constructive suggestions.

We had provided point-by-point response.

Major comments:

This is generally an informative study and the manuscript is well written. I have one major comment.  

  1. Subjective responses are integral to the interpretation of the study. This is evidenced by the focus on these outcomes in the introduction and discussion. Whilst I appreciate that the subjective outcomes were not the primary focus of the investigation, nevertheless, these assessed subjective outcomes warrant greater focus than is currently present. Please add details of what the specific end-points are and how these end-points were assessed to the methods. In addition, please provide the full data set in the results (i.e. figure 3). This includes the breakfast (control), FMD and water-only trials.

We have expanded the Figure 3 to include all 4 parameters for the satiety measure.

We have included the 7-item the food flavor/quality evaluation and the 10-item self-reported mood evaluation in the supplemental material (Figure S1 and S2). We have added a discussion on the limits of the study that the mood questionnaire may not have the sensitivity to detect the difference. LINE 295.

The Study Design section regarding the self-evaluation had been updated. LINE 107.

I have also provided several line-by-line minor comments that I hope the authors find useful.

Thank the reviewer’s kind effort. We had provided point-by-point response.

Minor comments

Abstract:

Line 24: Add a p-value for this effect

We added the p-value for the Glucose AUC0-4 comparison between FMD and water-fast groups. LINE 24.

Reframing of the study as water vs FMB is better.

We appreciate the reviewer’s original suggestion and have changed accordingly.

Introduction:

Line 44-49: This is a very long sentence. Consider splitting this into two separate sentences.

We have revised the sentences to reflect brevity. LINE 44, 47.

Line 68: Given the focus on appetite and compensatory eating discussed in the previous paragraph, this should be highlighted as a study outcome in this paragraph. This is an important part of the rationale for the study.

We have followed the recommendation of reviewers to highlight this point. LINE 67-70.

Methods:

Line 88: This line is confusing as the it mentions the breakfast condition in the randomisation schedule, as this study has been altered from a three trial to a two trial study. As such, it may be sensible to include lines 92-100 earlier, so that the purpose of the breakfast condition is clear to the reader.

We have edited the text according to the reviewer’s recommendation. LINE 90.

Line 109: I’m not sure why ‘or’ is used in this sentence. Consider revising.

It was a typo. It should be “for”. LINE 115.

Table 1: Please include the grams for each macronutrient (fat, protein, carbohydrate) in the table.

Table 1: The control breakfast should be included in this table.

We have included the information about breakfast and updated the Table 1 according to reviewer’s suggestion.

Line 120: How was the power calculation performed? Was this related to the FMD vs. water comparison or the FMD vs. control breakfast comparison?

The power calculation was performed based on FMD vs. control breakfast comparison in pilot studies. We have updated the description in the text. LINE 125.

Line 133-137: The measurement of these individual factors is not included in the manuscript. For example, were weight and height self-reported or independently measured by researchers? In particular, how physical activity level was determined is not explained. Please include how all of these individual factors were measured in the manuscript.

The height and weight measures were self-reported with the weight being measured on a study provided scale). All study participants completed demographics, medical history, concomitant medications and physical activity, drinking habits questionnaires modeled on the Nurses’ Health Study questionnaires.

We have included the description in the revised manuscript. LINE 107.

Line 150: Reword this sentence to ‘outnumbered males more than 2-fold’

We have updated the text accordingly. LINE 156.

Line 171-188: Please add units here, as appropriate. It may be more efficient to describe in the statistical analysis section of the methods that data is presented as mean (sd), to avoid the need to report (SD =) in each case here.

We have updated the text accordingly. LINE 178-191.

Line 190-191: Check this sentence for errors.

We have updated the text. LINE 197.

Line 198-204: These endpoints are not fully detailed in the methods, but are important for the interpretation of the study. This is confusing as the importance of appetite regulation in the introduction, yet the details pertaining to this in the methods and results of the manuscript are minimal. This is in stark contrast to the focus on (fairly predictable) effects of a very-low carbohydrate snack on glucose and BHB concentrations. Please add these details to the methods.

We have updated the method section. LINE107.

Figure 3: Present both baseline and 3-hour data points in this figure to observe the effect more clearly. It might also be interesting to include the breakfast meal in this figure for context. Add all the subjective outcomes to this figure. Details on exactly how these end-points were assessed (scale, anchors etc.) is needed in the methods.

We have updated the Figure 3 to include all 4 parameters for the motivation-to-eat analysis.

Discussion:

Line 230: The low carbohydrate content of the FMD bar is the major factor driving the observed effects and this should be highlighted in this sentence.

We have updated the text. LINE 238.

Line 224-237: Where have you derived the criteria for the descriptors? What constitutes high, moderate, low and very low in regard to macronutrient composition?  

The Classic ketogenic diet consists of a 4:1 ratio (in grams) of fat to carbohydrate and protein (e.g. KetoCal), although 3:1 and lower ratio had been used as well (Kossoff EH, McGrogan JR, 2005).

The Cleveland Clinic has defined the “The keto diet is essentially a high-fat diet — your meals are 70 or 80% fat; about 20% protein; and about 5% carbohydrate.” (https://health.clevelandclinic.org/what-is-the-keto-diet-and-should-you-try-it/)

We have added a reference in the text (Kossoff EH, McGrogan JR. PubMed PMID: 15679509.). LINE 244.

Line 237-239: This is very selective and focuses on the possible negative effects of protein. For balance, the potential positive effects of protein should also be highlighted here. In addition, are these mechanistic effects due to acute protein restriction? What would happen with subsequent feeding?  

We have added reference on protein restriction on human circulating IGF-1 level and more discussion. LINE 247 and LINE 250.

Line 239-242: The value stated for glucose is very specific and unlikely to be true for every person. The type of carbohydrate provided may also influence this response. Linking to the previous sentence, this sentence is designed to explain how carbohydrate provision reduces muscle-derived gluconeogenesis from amino acids, which could also be achieved via protein provision in the diet. These sentences read very selective and convoluted, and need consider the whole diet and wider implications of such an intervention.

We appreciate reviewer’s comments and had provided additional reference. We had emphasized the FMD is mimicking a short-term fast, rather than a chronic protein restriction. LINE 251. This section intends to contrast the FMD against the ketogenic diet. It provides the rationale of low protein, moderate carbohydrate nature of the FMD compared to the extremely low carb content of the ketogenic diet.

Line 249-254: This sentence is very good and identifies clearly the application of your findings. A major component of this is the potential to consume food and avoid some of the challenges associated with fasting. This needs to be a greater focus throughout the manuscript.

We appreciate the reviewer’s comments.

Line 280-283: Whilst I agree, a more detailed hourly subjective profile would be very beneficial to track the temporal response, this sentence is contradictory as it is stated that FMD is a snack, yet could be viewed as a healthy alternative to skipping breakfast (a meal). In this regard, if the FMD bar is designed to completely replace breakfast, the 3-hour time interval for the measurement of subjective responses is relevant, as practitioners would be expected to maintain the fast for the same duration as if eating breakfast.

We appreciate that the reviewer agrees with our study design of choosing the 3-hour time point. Meanwhile, we’d like to point out to readers the limitation of using 3-hour time point in this study for the classic satiety tests.

Reviewer 2 Report

Let's be clear: I am not going to buy a fanciful formulation that does better than a regular breakfast in prolonging the beneficial effects of FMD, given that such formulation provides less calories and has a different macronutrient composition that a regular breakfast. Of course the amount of calories provided is relevant! How can you say your control is only the water group?

My point is very clear, and it has not been addressed. 

Author Response

We deeply regret reviewer’s “to buy a fanciful formulation” comment.

More Americans are following a diet in 2020 than in 2019 (https://foodinsight.org/2020-food-and-health-survey/), with intermittent fasting edged out clean eating as the most common diet followed. We believe novel formulations and dietary programs that fasting mimicking concept could facilitate the fasting by provided not only moderate energy and nutrients but also behavioral and psychological support for the IF practitioners.

In the currently manuscript, we have indeed taken into account the multiple reviewers’ advice to focus on the non-inferiority comparison between FMD and water-fast. We have revised our manuscript based on reviewers’ comments to clarify that the energy level, the macronutrient type and ratio, timing are all relevant to implement a dietary program to mimic and, thus, facilitate the intermittent fasting. Both the water fast and breakfast groups matter in this study in that water fast serves as an active comparator for noninferiority analysis, and the breakfast group serves as the additional control to ensure that we observe the “break” of fast (as suggested by the name break-fast) and the adequacy of the glucose/ketone instrument. The novelty of the formulation rests with the fact that despite its calorie and macronutrient content, it achieves a similar physiologic state as water fast, in contrast to reaching glucose/ketone levels that are in between a water-fast and breakfast.

Reviewer 3 Report

Acceptable

Author Response

We thank the reviewer’s supportive suggestions and comments, and appreciate the reviewer’s approval our revised manuscript.

Round 2

Reviewer 1 Report

Thank you for addressing my comments on this manuscript. I have nothing further to add.